# Contact Lenses Loaded with Melatonin Analogs: A Promising Therapeutic Tool against Dry Eye Disease

**DOI:** 10.3390/jcm11123483

**Published:** 2022-06-17

**Authors:** Francisco Javier Navarro-Gil, Fernando Huete-Toral, Carmen Olalla Domínguez-Godínez, Gonzalo Carracedo, Almudena Crooke

**Affiliations:** 1Department of Optometry and Vision, Faculty of Optics and Optometry, Complutense University of Madrid, 28037 Madrid, Spain; codgeuo@opt.ucm.es (C.O.D.-G.); jgcarrac@ucm.es (G.C.); 2Department of Biochemistry and Molecular Biology, Faculty of Optics and Optometry, Complutense University of Madrid, 28037 Madrid, Spain; fhueteto@ucm.es

**Keywords:** drug delivery, contact lenses, 5-methoxycarbonylamino-N-cetyltryptamine, agomelatine, dry eye

## Abstract

Melatonin analogs topically administered evoke a potent tear secretagogue effect in rabbits. This route of drug administration requires high drug concentration and frequent dosing due to its reduced ocular surface retention. Therefore, contact lenses (CLs) have emerged as an alternative drug-delivery system that prolongs drug retention in the cornea, improving its therapeutic performance. This study explores the in vitro ability of five commercially available hydrogel CLs to act as a delivery system for melatonin analogs and the in vivo secretagogue effect of melatonin analog-loaded CLs. We soaked CLs with melatonin or melatonin analog solutions (1 mM) for 12 h. Spectroscopic assays showed that IIK7-loaded CLs led to the inadequate delivery of this compound. Conventional hydrogel lenses loaded with agomelatine released more agomelatine than silicone ones (16–33% more). In contrast, the CLs of silicone materials are more effective as a delivery system of 5-MCA-NAT than CLs of conventional materials (24–29%). The adaptation of CLs loaded with agomelatine or 5-MCA-NAT in rabbits triggered a higher tear secretion than the corresponding eye drops (78% and 59% more, respectively). These data suggest that CLs preloaded with melatonin analogs could be an adequate strategy to combat aqueous tear deficient dry eye disease.

## 1. Introduction

Dry eye disease (DED) is a common disabling disorder that affects the patients’ visual function and quality of life [1,2]. Given that age is a pivotal risk factor for DED, the current aging population, and its marked socioeconomic impact, this disorder is a growing public health concern [1,2,3]. DED is a multifactorial disease of the ocular surface in which tear film instability, hyperosmolarity, inflammation, and oxidative stress play an etiological role [4,5]. As a result of these intricate molecular interactions underlying DED, the therapy approach is complex and frequently unable to provide fast and complete symptom alleviation [6,7,8,9]. Therefore, researchers are trying to develop effective interventions to restore ocular surface health.

Secretagogues are one of the most prominent agents to treat aqueous deficient DED [10,11]. These molecules encourage tears, mucin, and lipid secretion, improving tear film stability and therefore this condition. Currently, there are two approved secretagogues in the Asian market [Diquas^®^ (diquafosol tetrasodium, Up_4_U), and Mucosta^®^ (rebamipide)], and several molecules are under development [9,12]. Melatonin analogs are an example of this last group, triggering prominent tear stimulation in rabbits [13]. These analogs interact with melatonin membrane receptors, which are present in various eye structures such as the ocular surface and lacrimal gland, mimicking melatonin eye effects [14,15,16,17]. Researchers have also developed these analogs to overcome the poor pharmacokinetic properties and low subtype receptor selectivity of melatonin [18,19]. Given that membrane melatonin receptors are present throughout the whole eye, some authors have claimed the therapeutic potential of melatonin and its analogs against several eye diseases including DED [14,15,19,20,21,22]. Melatonin, via activation of its membrane receptors, can regulate ocular physiological processes such as the regulation of retinomotor movements, rod outer segment disc shedding, and humor aqueous secretion, which is pivotal for intraocular pressure modulation [23,24,25,26]. Concerning DED damage, melatonin can modulate epithelial wound healing on the ocular surface and potentiate the secretagogue effect of another experimental secretory stimulant, Ap_4_A [15,27]. Likewise, melatonin may improve hyperosmolarity, inflammation, and oxidative stress in animal models of DED [21,22]. Consequently, melatonin analogs could be an adequate strategy to combat DED for their secretagogue effects and other protective actions essential to restoring ocular surface homeostasis under this condition.

Secretagogues are mainly administered topically in the eye [9]. This route of administration has a low corneal bioavailability, requiring high drug concentration and recurrent dosing to achieve the therapeutic effect [28,29]. Therefore, it increases the risks of side effects and causes poor patient compliance [29]. Drug delivery by contact lenses (CLs) has emerged as a pharmaceutical alternative to topical eye formulations [30,31]. Indeed, the drug released from CLs can present a corneal bioavailability of 50% (10-fold higher than by topical administration) [31].

There are different methods to incorporate the drug into the lens [30,31]. One of these methods is soaking, which is the easiest, fastest, and cheapest [30]. Conversely, it has some disadvantages (e.g., low drug loading and quick drug release), but it has allowed for the inclusion of numerous ophthalmologic drugs into the lenses and their appropriate release [30,31].

Several protocols also exist to determine the in vitro drug release profile, leading in many cases to false conclusions that do not necessarily replicate under in vivo conditions [32]. Nevertheless, in vitro studies are critical to a quick and easy screening of molecules with the ability to penetrate the contact lens and exit from it adequately, minimizing the number of animals needed to perform the in vivo studies.

Therefore, this study aimed to explore the in vitro ability of five commercially available hydrogel CLs to incorporate melatonin analogs by soaking and releasing them effectively. Moreover, we tested the in vivo secretagogue effect of melatonin analog-soaked CLs and compared it with the effect triggers via the topical administration of these molecules. Previous works have suggested the use of a platform for the ocular drug delivery of only two of the five lenses tested in our study [30].

## 2. Materials and Methods

### 2.1. Materials

Melatonin, 5-methoxycarbonylamino-N-acetyltryptamine (5MCA-NAT), and N-butanoyl-2-(9-methoxy-6H-isoindolo [2,1-a]indol-11-yl) ethanamine (IIK7) were purchased from Tocris (Bristol, UK) and agomelatine was purchased from Santa Cruz Biotechnology Inc. (Dallas, TX, USA). All of the compounds were formulated in isotonic saline containing 1% DMSO from Sigma-Aldrich (St. Louis, MO, USA). Three commercially available silicone hydrogel (SH) contact lens (CL) materials [Balafilcon A (PureVision 2, Bausch & Lomb, Bridgewater, NJ, USA), Comfilcon A (Biofinity, CooperVision, San Ramon, CA, USA), and Stenfilcon-A (MyDay, CooperVision)], and two conventional hydrogel (CH) CL materials [Omafilcon A (Proclear, CooperVision) and poly(2-hydroxyethyl methacrylate), p-HEMA, (Veraflex T, Interlenco, Spain)] were evaluated in the study. Table 1 details the properties of these SH and CH CLs. All lenses were obtained from the manufacturer in the original and each lens had three dioptric powers (±2.00 and +0.00 or −0.25).

### 2.2. Animals

Male New Zealand white rabbits weighing 3 to 4 kg were placed in individual cages with free access to food and water and subjected to regular light/dark cycles (12 h). All experiments were performed according to the Association for Research in Vision and Ophthalmology Statement for the Use of Animals in Ophthalmic and Vision Research and to the European Directive 2010/63/EU.

### 2.3. Spectrophotometric Determination of Melatonin and Its Analogs

The absorbance spectra of melatonin, 5-MCA-NAT, IIK7, and agomelatine (1 mM) were determined at 25 °C using a 96-well quartz microplate (Hellma Analytics GmbH, Müllheim, Germany) and a Power Wave XS2 Microplate Spectrophotometer (BioTek Instruments Inc., Winooski, VT, USA) between 200 and 500 nm. None of them presented an absorption above 370 nm. The spectra had an absorption maximum for melatonin and 5-MCA-NAT corresponding to the 230 nm wavelength. Likewise, the maximum absorbance for IIK7 and agomelatine were at 220 nm and 275 nm, respectively. Thus, these wavelengths (230 nm, 220 nm, and 275 nm) were used to detect melatonin and its analogs in drug release and uptake studies.

### 2.4. Preparation of Drug-Loaded CL and In Vitro Studies of CL Drug Release

The CLs were removed from the blisters and then dried on blotting paper. Subsequently, CLs were placed on the wells of a 24-well plate and soaked in 1 mL of each drug-DMSO solution (1 mM of melatonin and its analogs) for 12 h at room temperature. After that, the lenses were taken out and the superficial drug solution was removed by rubbing the lens against the edges of the well. CLs were then placed in new individual wells with a volume of 1 mL of saline solution and 100 μL of the resulting solutions (saline solution containing the released drug) were removed at 5, 15, 30, 60, 120, 180, 240, and 360 min. Finally, the CLs were taken out and placed in new wells with 1 mL of fresh saline solution overnight. This drug release time course was then repeated for another 360 min. The concentration of melatonin/analogs released was determined by spectrophotometry at their specific wavelengths (230 nm, 220 nm, and 275 nm) using a 96-well quartz microplate and a Power Wave XS2 Microplate Spectrophotometer. After the absorbance measurements, all 100 μL aliquots were pipetted back into their original well. To generate a linear calibration curve and thus correlate the absorbance readings to concentrations, the stock solutions of melatonin and its analogs were diluted to a range between 0.001 mM and 1 mM in 1% DMSO.

### 2.5. In Vitro Studies of CL Drug Uptake

The contact lenses were removed from their containers and then dried on blotting paper. After that, the CLs were placed on wells of a 24-well plate and soaked in 1 mL of 5-MCA-NAT or agomelatine 1 mM in 1% DMSO at room temperature. Aliquots (50 μL) were removed from the source solution at 5, 15, 30, 60, 120, 180, 240, and 360 min to assay the drug remaining in the solution by spectrophotometry at 230 nm (for 5-MCA-NAT) and 275 nm (for agomelatine) using a 96-well quartz microplate and a Power Wave XS2 Microplate Spectrophotometer. We performed 5-MCA-NAT and agomelatine linear calibration curves (curves ranging between 0.001 mM and 1 mM in 1% DMSO) to correlate the absorbance reading with their respective concentration.

### 2.6. In Vivo Studies of the Secretagogue Activity of Melatonin Analogs Pre-Soaked CLs

The CLs were pre-loaded with agomelatine (100 μM; p-HEMA CLs) and 5-MCA-NAT (250 μM, Stenfilcon-A CLs) for 12 h and after removal of the superficial drug solution and carefully placed on the cornea of rabbits (one cornea per rabbit). After 5 min of measuring the basal tear level and adaptation of the lenses, tear fluids were collected from the inferior temporal eyelid at 5, 15, 30, 60, 120, 180, and 240 min using Schirmer’s strips (Whatman, Buckinghamshire, UK) and the protocol described by Van Bjisterveld [33].

### 2.7. In Vivo Studies of the Secretagogue Activity of Melatonin Analogs via Topical Administration

Agomelatine and 5-MCA-NAT were instilled at a volume of 10 μL at 100 μM and 250 μM, respectively. The control experiments were performed by applying 10 μL of saline solution containing 1% DMSO, 15 min before any of the other compounds/CLs were used. After 5 min of the saline application (controls) or the adaptation of loaded lenses/instillation of analogs, tear fluids were collected from the inferior temporal eyelid at 5, 15, and 120 min using Schirmer’s strips (Whatman, Buckinghamshire, UK) and the protocol described by Van Bjisterveld [33].

### 2.8. Statistical Analysis

Statistical analysis was performed using SSPS Statistic 23 software (IBM, Chicago, IL, USA). The normal distribution of variables was evaluated with the Shapiro–Wilks test. Differences between groups in the CL drug uptake/release studies were determined using a Student’s unpaired *t*-test and one-way ANOVA, which was followed by a post hoc rank test (Bonferroni) when multiple comparisons were performed. The Student’s unpaired *t*-test was also used to evaluate the secretagogue effect induced by the adaptation of melatonin analog-loaded CLs on rabbits and the eye instillation of these analogs. In all cases, values were considered statistically significant at *p* < 0.05. The adequacy of the sample size for the in vivo studies (*n* = 6) was tested using the statistical software Granmo 6.0 (Institut Municipal d’Investigació Médica, Barcelona, Spain) with an accepted two-sided alpha risk of 0.05, a beta risk of 0.20, an estimated common standard deviation of 0.6 units, and a minimum expected difference of 1.0 units. Graphs were plotted using GraphPad Prism 6 software (GraphPad Software, La Jolla, CA, USA).

## 3. Results

### 3.1. Release of Melatonin and Its Analogs from CLs

The released concentration of each melatonin compound from the same kind of CL and for each dioptric power at any of the experimental times studied was similar, ruling out any effect of dioptric power on their delivery (*n* = 3 CLs/dioptric power, *p* > 0.05 ANOVA) (data not shown). Therefore, CLs of any dioptric power (±2.00, +0.00, or −0.25) were used for the remaining in vitro and in vivo experiments.

Among all of the CLs pre-soaked with melatonin and its analogs, the IIK7-loaded ones showed the worst ability to deliver these molecules. CLs with Omafilcon B and Comfilcon A materials released at 15 min the maximum concentration of IIK7, which was 70.07 ± 2.94 µM S.D. and 67.9 ± 12.33 µM S.D., respectively (*p* < 0.05, ANOVA) (Figure 1). Nonetheless, the release of IIK7 from CLs lasted a short time (maximum duration 30 min with IIk7-loaded p-HEMA) (Figure 1).

Regarding melatonin, it reached the maximum significant concentration values at 15 min with the CH materials, p-HEMA (147.79 ± 12.62 µM S.D.) and Omafilcon B (154.66 ± 15.59 µM S.D.), and with the SH materials Comfilcon A (143.84 ± 15.62 µM S.D.) and Stenfilcon A (146.09 ± 12.61 µM S.D.) (*p* < 0.05, ANOVA), without statistically significant differences between both families of materials (Figure 2). In the case of Balafilcon A CLs, the maximum release of melatonin occurs at 60 min (156.51 ± 15.91 µM S.D., *p* < 0.05, ANOVA) (Figure 2). This event may be due to the ionic character and the surface coating of Balafilcon A CL, which could promote the interaction between melatonin and CL, complicating its release.

Figure 3 shows the agomelatine released from pre-soaked CLs. CH CLs delivered a higher concentration of this melatonin analog than the SH ones. Therefore, p-HEMA CLs released a maximum concentration of 166.60 ± 3.97 µM S.D at 60 min (*p* < 0.05, ANOVA) (Figure 3). Likewise, Omafilcon B CLs released a maximum concentration of 151.07 ± 14.43 µM S.D. (*p* < 0.05, ANOVA), although this was achieved later (at 120 min) (Figure 3). The quicker release from p-HEMA may be due to its lower water content than Omafilcon B lenses (WC% p-HEMA = 38%; WC% Omafilcon B = 62%) (Table 1).

Regarding SH CLs, Comfilcon A released a maximum of 140.43 ± 17.45 µM S.D at 30 min (Figure 3). Similarly, Stenfilcon A lenses delivered a maximum level of agomelatine of 138.25 ± 12.37 µM S.D., although it occurred later (at 180 min) (Figure 3). In contrast, Balafilcon A CLs delivered the lowest maximal concentration at 240 min (101.04 ± 11.52 µM S.D.) (Figure 3). Given that the water content of Balafilcon A CL is similar to that of CH p-HEMA (36% versus 38%, respectively), it does not seem crucial to regulate the agomelatine release from this lens. In contrast, the ionic character and the surface coating, absent in the other lenses, could be determinants.

Given the high ability to deliver agomelatine from the loaded CLs, we repeated the measurement of its release after overnight CL incubation in the release solution and the renewal of the saline medium. Like in the first measures, p-HEMA and Omafilcon B delivered the maximum concentration of agomelatine (170.23 ± 7.5 µM S.D and 148.52 ± 0.56 µM S.D., respectively, and at 360 min; *p* < 0.05 ANOVA) (Figure 3).

Unlike agomelatine, the 5-MCA-NAT release was easier from SH CLs than from the CH (Figure 4). Stenfilcon A lenses delivered the maximum level of 5-MCA-NAT at 240 min (805.6 ± 24.67 µM S.D., *p* < 0.05, ANOVA) (Figure 4). At the same time, Comfilcon A lenses released a maximum concentration of 747.89 ± 32.24 µM S.D. (*p* < 0.05, ANOVA) (Figure 4). The maximum delivery from Balafilcon A occurred quicker (at 180 min) than from the other SH lenses (728.95 ± 36.04 µM S.D., *p* < 0.05, ANOVA) (Figure 4). Once more, the unusual release behavior of this lens could be due to its ionic character and surface coating. Conversely, the CH lenses p-HEMA and Omafilcon B released the lowest maximum concentration of 5-MCA-NAT (521.38 ± 69.58 µM S.D. and 591.38 ± 15.46 µM S.D., respectively, and at 120 min, *p* < 0.05, ANOVA) (Figure 4).

As a result of the high concentration of the 5-MCA-NAT outed, we repeated the release assay after overnight CL incubation in fresh saline solution and the renewal of this solution. These second measures confirmed the easier release of 5-MCA-NAT from the SH lenses, although the concentration values obtained were significantly inferior to the values obtained in the first release assay. Therefore, the maximum concentration obtained from Comfilcon A was 130.19 ± 4.31 µM S.D.; from Stenfilcon A, it was 133.69 ± 7.9 µM S.D. (both obtained at 240 min); and from Balafilcon A it was 99.26 ± 14.27 µM S.D. (this value was obtained at 540 min) (Figure 4).

### 3.2. Uptake of Agomelatine and 5-MCA-NAT by CLs

The release results obtained with agomelatine and 5-MCA-NAT-loaded CLs suggest that CLs could be an adequate system for their eye delivery. Nevertheless, to evaluate the yield of CLs as a drug delivery system, it is also necessary to know the concentration of drug uptake by the lens (Phan et al., 2018). Therefore, we analyzed the concentration of agomelatine and 5-MCA-NAT uptake by lenses in 1 mL of the corresponding melatonin analog solution (1 mM) (Figure 5 and Figure 6).

Regarding agomelatine, and unlike the release, the SH CLs absorbed its maximum concentration (*p* < 0.05, ANOVA) (Figure 5). Therefore, Stenfilcon A lenses absorb 906.44 ± 19.56 µM S.D., Comfilcon A CLs absorb 884.7 ± 41.42 µM S.D., and Balafilcon A CLs absorb 895.43 ± 11.45 µM S.D, all at 120 min. The absorbed concentration did not vary beyond this point (Figure 5). With the SH lenses, the absorption did not seem to be affected by their ionic character and surface treatment as the absorption values of the Balafilcon A lenses were similar to the rest of the SH CLs.

Figure 5 also shows that the CH lenses absorbed a lower maximum agomelatine concentration and did it slower than the SH ones. Moreover, none of the CH lenses were saturated with agomelatine at the end of the experiment (Figure 5). Omafilcon A lenses absorbed a maximum of 691.67 ± 19.64 µM S.D and the p-HEMA ones 613.97 ± 21.36 µM S.D., at 300 min (*p* < 0.05, ANOVA) (Figure 5). Absorption by the CH CLs seemed to be affected by water content, being higher in the lenses with a higher content of Omalficon A. Conversely, the ratio between the maximum absorbed concentration and the maximum released concentration of agomelatine gave the yield of agomelatine-loaded CLs as a drug-delivery system (see Table 2). As Table 2 shows, the p-HEMA lenses provided the best ratio. Therefore, we used these CH lenses to perform the in vivo assays.

The uptake assay with 5-MCA-NAT showed that the maximum absorption occurred with Stenfilcon A (681.44 ± 23.68 µM S.D.), followed by Comfilcon A (648.00 ± 38.21 µM S.D.) and the CH p-HEMA (507.98 ± 20.14 µM S.D.) (*p* < 0.05, ANOVA) (Figure 6). The maximum absorption with the SH Balafilcon A lenses was 397.04 ± 15.86 µM S.D. and with CH Omafilcon B CLs was 407.55 ± 26.88 µM S.D., values considerably lower than those obtained for the rest of the lenses (*p* < 0.05, ANOVA) (Figure 6). These results suggest that neither the water content, the surface charger, surface treatment, nor the material affected the 5-MCA-NAT absorption. Moreover, none of the lenses were saturated with 5-MCA-NAT at the end of the experiment (Figure 6). Table 3 shows the ratio between the maximum absorbed concentration and the maximum released concentration of 5-MCA-NAT of each lens. In this case, Balafilcon A CLs provided the worst yield as a drug delivery system (Table 3). The rest of the lenses procured a similar yield (close to 1:1) (Table 3). Therefore, we used Stenfilcon A lenses to perform the in vivo assays.

### 3.3. Effect of Melatonin Analogs Released from CLs and Administered Topically in Rabbit Tear Secretion

To explore the effect of preloaded-CLs in rabbit tear secretion, we soaked p-HEMA lenses with a solution of 100 μM agomelatine for 12 h. This concentration evokes the maximum secretagogue effect after its topical administration (Navarro Gil et al., 2019) [13].

The adaptation of agomelatine-preloaded CLs in rabbits triggered a significant increase in tear secretion after 15 min in comparison with the control group (185.8 ± 14.25% S.E.M., *p* < 0.05 Student’s *t*-test) (Figure 7). Moreover, the maximum increase occurred after 180 min of the adaptation (216.9 ± 37.03% S.E.M.; *p* < 0.05 Student’s *t*-test), and subsequently, it was reduced to 209.8 ± 32.15% S.E.M. at 240 min; (*p* < 0.05 Student’s *t*-test) (Figure 7). The topical administration also evoked a significant increase in the tear secretion after 15 min (137.5 ± 14.25% S.E.M., *p* < 0.05 Student’s *t*-test) (Figure 7). The maximum tear secretion increase occurred after 60 min of instillation (138.9 ± 6.5% S.E.M., *p* < 0.05 Student’s *t*-test), and was subsequently reduced to 128.0 ± 5.7% S.E.M at 120 min (*p* < 0.05 Student’s *t*-test) (Figure 7). These data indicate that the adaptation of agomelatine-loaded p-HEMA CLs in rabbits triggers a higher tear secretion than the topical eye administration of agomelatine (78% more than the maximal tear secretion increase in its topical administration) (Figure 7).

To evaluate the tear secretagogue effect triggered with CLs loaded with 5-MCA-NAT, we soaked the Stenfilcon A lenses with a solution of 5-MCA-NAT 250 μM for 12 h. This concentration evoked the maximum secretagogue effect after its topical administration (Navarro Gil et al., 2019) [13]. The adaptation of 5-MCA-NAT-preloaded CLs in rabbits triggered a significant increase in tear secretion after 15 min in comparison with the control group (170.9 ± 18.31% S.E.M., *p* < 0.05 Student’s *t*-test), and a maximum increase in the tear secretion at 120 min (178.4 ± 19.71% S.E.M.; *p* < 0.05 Student’s *t*-test) (Figure 8). This tear secretion increase was maintained after 240 min (159.0 ± 12.94% S.E.M., *p* < 0.05 Student’s *t*-test) (Figure 8). The topical administration also evoked a significant increase in tear secretion after 15 min (118.65 ± 4.6% S.E.M., *p* < 0.05 Student’s *t*-test) (Figure 7). The maximum tear secretion increase occurred after 60 min of instillation (119.35 ± 4.6% S.E.M., *p* < 0.05 Student’s *t*-test), which was maintained up to 120 min (Figure 8). These data indicate that the adaptation of 5-MCA-NAT-loaded Stenfilcon A CLs in rabbits triggers a higher tear secretion than the topical eye administration of 5-MCA-NAT (59% more than the maximal tear secretion increase in its topical administration) (Figure 8).

The present findings confirm that melatonin analog-soaked CLs are effective drug-delivery systems of these analogs in the cornea. Moreover, our results showed that melatonin analog-loaded CLs triggered a potent and higher tear secretagogue effect than melatonin analog eye drops.

## 4. Discussion

Secretagogues are one of the most prominent agents to treat aqueous deficient dry eye disease [10,11]. These molecules encourage mucin and tear secretion, improving tear film stability and therefore this condition. In this context, we previously reported that the topical eye administration of melatonin analogs promotes tear secretion in rabbits [13].

In the eye, topical drug administration requires a high drug concentration and frequent dosing due to its reduced ocular surface retention [29]. Consequently, it increases the risks of side effects and leads to poor patient compliance [29]. Therefore, drug-loaded contact lenses have emerged as an alternative to topical eye formulations to prolong drug retention on the cornea [30,31].

In our present study, we analyzed the in vitro ability of commercially available conventional and silicone hydrogel CLs to act as a delivery system for melatonin analogs. Moreover, we explored the in vivo secretagogue effect of melatonin analog-loaded CLs and compared it with the effect triggered via the topical administration of these molecules.

Among the different existing methods to integrate a drug into a CL, we employed the most manageable and cost-effective, the soaking method [30,31]. This method has allowed for the adequate incorporation of numerous ophthalmologic drugs into the lenses and their release [31].

Various authors have demonstrated that the efficacy of soaked-CLs as a drug delivery system depends on the physicochemical compatibility between the CL and the drug [31,34,35,36]. In this sense, the characteristics of the CLs (e.g., water content and ionic charge) and drug properties (e.g., lipophilicity and molecular weight) are pivotal in the drug uptake and release [31,37,38].

Our in vitro results suggest that the load and release of melatonin analogs depend on the same factors. Therefore, the water content of CLs has affected the concentration of agomelatine absorbed by the CH lenses and its release kinetics. Indeed, various studies have suggested that a low water content accelerates the delivery from CH lenses [31,37,39,40]. Likewise, the ionic charge and the surface treatment of the CLs altered the agomelatine release. Balafilcon A lenses, which were the only CLs tested with an ionic characteristic and treatment of its surface in our study, showed the lowest delivery of this melatonin analog. Regarding the surface coating of this lens, Read et al. demonstrated that it caused hydrophilic islands that could interact with the hydrophilic side of agomelatine [41]. Conversely, other authors have proven the ability of agomelatine to interact with ionic lipids [42]. Therefore, the amphipathic agomelatine could interact strongly with Balafilcon A, delaying its release. This fact confirms the importance of studying the properties of the drug to the design of a successful drug-delivery system from CLs. Thus, we have observed that whereas SH CLs provide agomelatine maximum uptake, CH CLs are more effective in the melatonin analog release. This fact could be due to the high affinity of the agomelatine for siloxane groups of SH lenses, which allows for a high load of the lenses, but prevents its release.

In the case of 5-MCA-NAT, this event did not occur, and the absorption and release were favored by SH lenses (only Balafilcon A compromised its absorption). This difference between the 5-MCA-NAT and agomelatine behavior could be due to their different lipophilicity. Indeed, 5-MCA-NAT had a lower XLogP3 value than agomelatine, being a more hydrophilic molecule [XLogP3 5-MCA-NAT = 0.4 versus XLogP3 agomelatine = 2.7; Computed by XLogP3 3.0 (PubChem release 7 May 2021)].

The characteristics of the melatonin analog IIK7 also influence its release, causing a low and brief release of this molecule from soaked CLs. This analog is the most lipophilic and the largest of the three analogs tested [XLogP3 = 3.7, computed by XLogP3 3.0 (PubChem release 7 May 2021), and 30% bigger than the other analogs]. These attributes could promote the IIK7 adherence to CLs and prevent its penetration. Indeed, some authors have demonstrated that the molecular weight and lipophilicity of the drug can modulate its entrance into the CLs [39,43].

Equally important for a successful drug-delivery system from CLs is that the concentration released from lenses in vitro would be enough to exert the in vivo desired effect. The concentration of melatonin released from CLs is far from the minimum effective dose that causes a secretagogue effect in rabbits (1 mM, determined by [13]). Unlike melatonin, all of its analogs tested in our study released a sufficient concentration of these compounds, stimulating tear secretion in rabbits. IIK7-loaded CLs released a low concentration of IIK7 (maximum concentration released ≈70 μM), but enough to evoke the half-maximal secretagogue effect in rabbits (EC_50_ IIK7 = 69.2 µM, via topical administration) [13]. Conversely, the agomelatine and 5-MCA-NAT-loaded lenses released sufficient melatonin analog to produce the maximum secretagogue effect (EC_50_ agomelatine = 52.1 µM and EC_50_ 5-MCA-NAT = 99.5 µM, via topical administration) [13]. Therefore, soaked CLs seem to be adequate to deliver these melatonin analogs.

Many studies have demonstrated that most in vitro tests are not good predictors of the in vivo efficacy of drug delivery from CLs [32]. This fact is mainly due to the absence of an officially approved protocol for testing in vitro drug release, which has promoted the use of a myriad of experimental conditions [32,44]. These conditions, in many cases, lead to false conclusions about the drug release from lenses that do not reproduce in the complex eye structure [32,44]. Therefore, researchers claim to standardize the in vitro drug release protocol and improve it to resemble more closely intricate tear dynamics [33,44]. While this fact occurs, it is necessary to verify the in vitro release results in the animal model’s eye before testing these drug-loaded CLs in the human eye. For this reason, we evaluated the correlation between the in vitro and in vivo efficacy of melatonin analog-loaded CLs as a drug-delivery system in rabbits (i.e., a system capable of releasing melatonin analogs and stimulating tear secretion). Likewise, we compared the secretagogue effect triggered by melatonin analog-loaded CLs versus the effect evoked by the topical administration of these analogs.

Our in vivo results showed that the adaptation of agomelatine-loaded p-HEMA CLs and 5-MCA-NAT-loaded Stenfilcon A CLs in rabbits triggered a marked increase in tear secretion compared with the control rabbits. Most importantly, these increases were higher than the increase evoked by the topical administration of melatonin analogs (78% and 59% more, respectively). Consequently, melatonin analog-loaded CLs are effective drug-delivery systems and better alternatives as secretagogue agents than melatonin analog eye drops in rabbits. As we commented above, the melatonin concentrations released by the lenses in vitro were far from the minimum effective dose that causes a secretagogue effect in rabbits after its topical administration. Given the more potent secretagogue effect obtained via the delivery of CLs versus the topical route, the melatonin released by soaked CLs could trigger that effect in vivo. Indeed, Serramito et al. recently demonstrated that the melatonin-loaded CLs produced by solvent casting evoked a marked secretagogue effect in rabbits [45]. This method of drug incorporation has emerged to overcome the main problems associated with the soaking protocol [30].

The secretagogue effect provided by lenses soaked with melatonin analogs is also more sustained than the effect obtained by Diquafosol tetrasodium (Up_4_U, a drug approved in Asiatic countries for the treatment of DED) [46,47]. The topical administration of this dinucleotide in eyes increased the tear meniscus height (in humans) and pre-lens and post-lens tear film (in rabbits) between 15 min and 60 min [46,47]. Nevertheless, Up_4_U, or the dinucleotide Ap_4_A, released from contact lenses, sustains this secretagogue effect over five hours [27,48]. Several works have demonstrated that Up_4_U stimulates tears, mucin, and lipid secretion and improves the morphology of impaired meibomian glands [49,50]. Given the high prevalence of meibomian gland dysfunction among DED patients, future studies should investigate the potential effect of melatonin analogs on lipid secretion [51].

Regardless, these data suggest that the employment of CLs as a drug-delivery system in the cornea could be a valuable tool for managing DED. Furthermore, given that the last CL prescribing data showed a dramatic increase in the fitting of daily disposable lenses, the soaked lenses, with a drug release over 5–6 h and are faster and cheaper than other kinds of loaded CLs, seem to be a better alternative [30,52].

Concerning a possible translation of these animal results to humans, it is necessary to consider that the rabbit eye presents a different tear film turnover and blinking occurrence than the human eye, impairing the drug release kinetics and residence in the human cornea [32]. Conversely, the rabbit presents a nictitating membrane that reduces the stability of the CL [53]. Therefore, although this animal model is commonly used to evaluate the drug release from CLs, it is necessary to take rabbit findings with caution.

Regardless, using CLs as a vehicle of melatonin analog delivery could reduce the dose of these secretagogue agents, avoiding possible adverse effects. Serramito et al. demonstrated, by using the Draize test, that melatonin at the same concentration ranked similar to the one employed in our in vivo studies (micromolar range) did not provoke corneal adverse effects in rabbits [45]. Furthermore, another study confirmed the good ocular tolerance of melatonin at the millimolar range of concentration in a different animal model [22].

We neither observed signs of eye damage after the adaptation of the lenses loaded with the melatonin analogs or the topical administration of these analogs. Nevertheless, this issue is particularly significant because agomelatine intake can induce a hepatotoxic effect [54]. Given the emerging therapeutic potential of melatonin and thus its analogs in eye diseases [20], the eye toxicity of these compounds should constitute the object of future studies.

Our results demonstrate that the agomelatine and 5-MCA-NAT released from lenses evoked a marked secretagogue effect. The tear film instability, hyperosmolarity, ocular surface inflammation, and oxidative stress are characteristic of DED [4,5]. Some authors have recently demonstrated that melatonin improved the hyperosmolarity, inflammation, and oxidative stress in animal models of DED. Therefore, they have suggested its therapeutic potential against DED [21,22]. Agomelatine presents anti-inflammatory and antioxidative activity [55,56,57], but it is unknown whether it has this effect on the ocular surface of DED animal models. Future investigations are necessary to validate the actions of agomelatine and 5-MCA-NAT analogs on the ocular surface in conditions that mimic DED.

## 5. Conclusions

The present findings confirm that soaked contact lenses are effective drug-delivery systems in the cornea and could be valuable tools for managing aqueous DED. Moreover, our results showed that agomelatine/5-MCA-NAT-loaded CLs triggered a potent and higher tear secretagogue effect than melatonin analog eye drops. Secretagogues are one of the most prominent agents to treat aqueous deficient dry eye disease because of the improvements in tear, mucin, and lipid secretion and thus tear film stability. Consequently, melatonin analog-loaded CLs could be an alternative to the current topical secretagogues against dry eye disease.

## Figures and Tables

**Figure 1 jcm-11-03483-f001:**
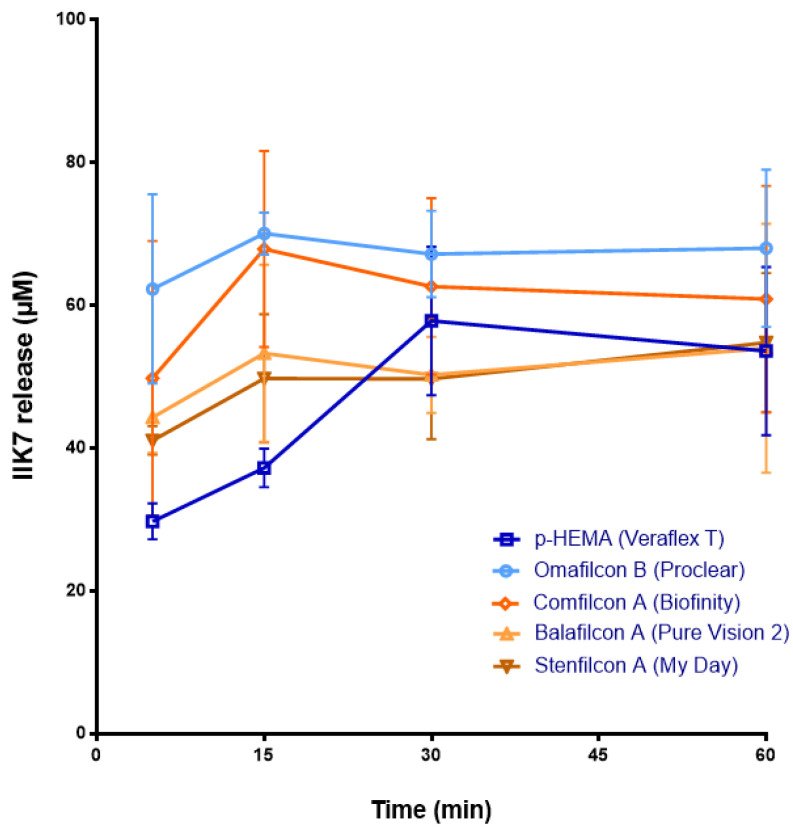
The IIK7 release from CLs previously soaked with IIK7 solution (1 mL, 1 mM) for 12 h to a saline solution (1 mL) over 60 min. Concentration (μM) values are presented as mean ± S.D of *n* = 9 lenses per material. The blue color shows the release from CH CLs. The ochre color shows the release from SH CLs.

**Figure 2 jcm-11-03483-f002:**
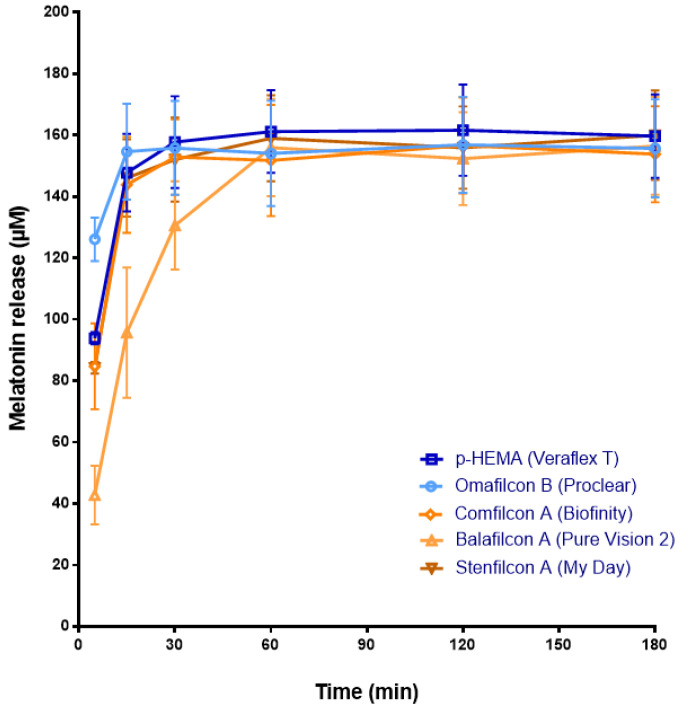
The melatonin release from CLs previously soaked with melatonin solution (1 mL, 1 mM) for 12 h to a saline solution (1 mL) over 180 min. Concentration (μM) values are presented as the mean ± S.D of *n* = 9 lenses per material. The blue color shows the release from CH CLs. The ochre color shows the release from SH CLs.

**Figure 3 jcm-11-03483-f003:**
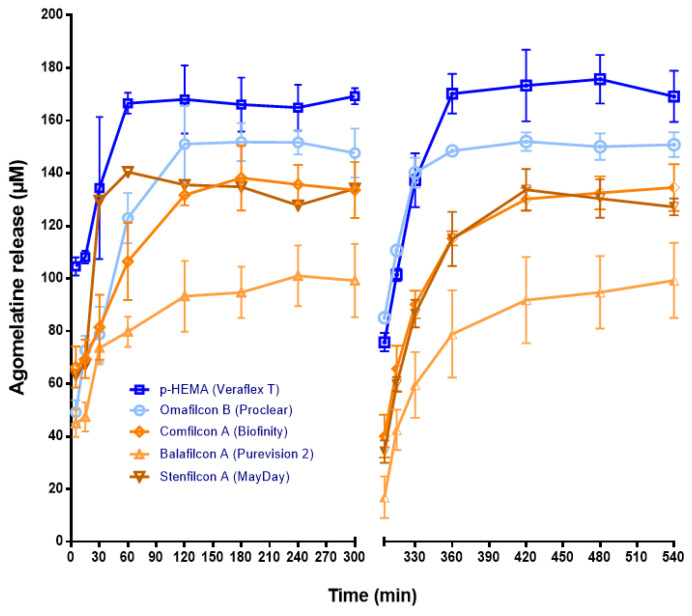
The agomelatine release from CLs previously soaked with agomelatine solution (1 mL, 1 mM) for 12 h to a saline solution (1 mL) over 540 min, with a change in the saline medium at 300 min. Concentration (μM) values are presented as mean ± S.D of *n* = 9 lenses per material. The blue color shows the release from CH CLs. The ochre color shows the release from SH CLs.

**Figure 4 jcm-11-03483-f004:**
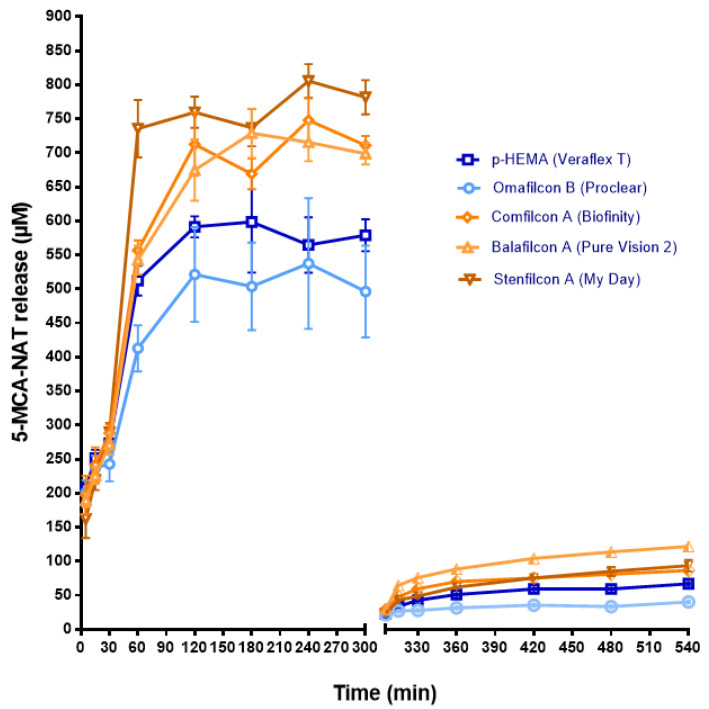
The 5-MCA-NAT release from CLs previously soaked with the 5-MCA-NAT solution (1 mL, 1 mM) for 12 h to a saline solution (1 mL) over 540 min, with a change in the saline medium at 300 min. The concentration (μM) values are presented as the mean ± S.D of *n* = 9 lenses per material. The blue color shows the release from CH CLs. The ochre color shows the release from SH CLs.

**Figure 5 jcm-11-03483-f005:**
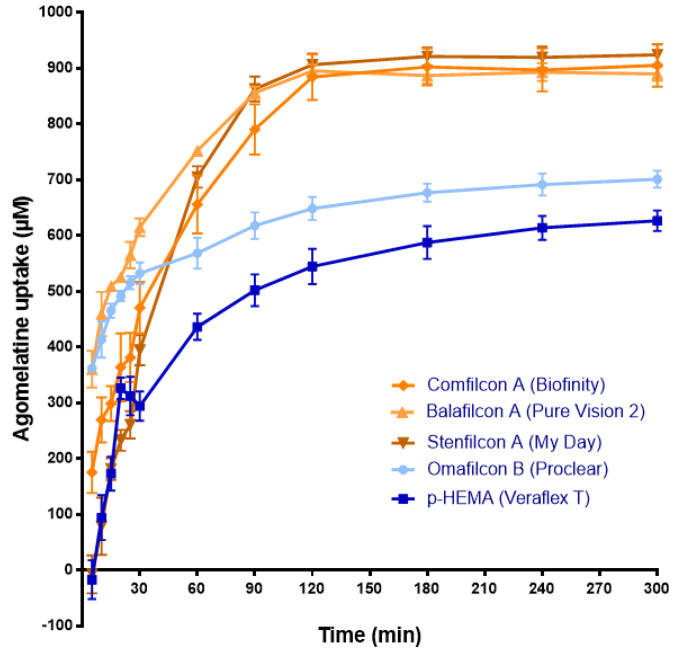
The agomelatine absorption by the CLs soaked with agomelatine solution (1 mL, 1 mM) over 300 min. The concentration (μM) values are presented as the mean ± S.D of *n* = 3 lenses per material. The blue color shows the release from CH CLs. The ochre color shows the release from SH CLs.

**Figure 6 jcm-11-03483-f006:**
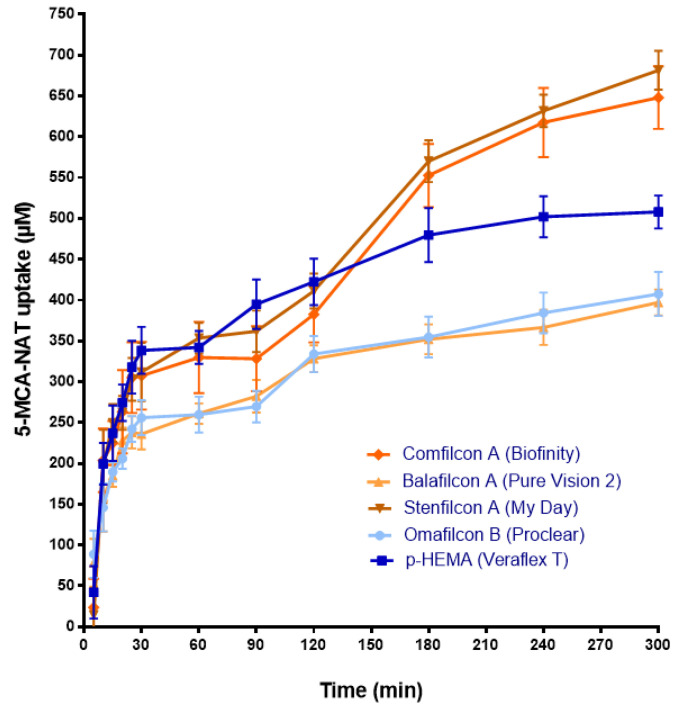
The 5-MCA-NAT absorption by CLs soaked with 5-MCA-NAT solution (1 mL, 1 mM) over 300 min. The concentration (μM) values are presented as the mean ± S.D of *n* = 3 lenses per material. The blue color shows the release from CH CLs. The ochre color shows the release from SH CLs.

**Figure 7 jcm-11-03483-f007:**
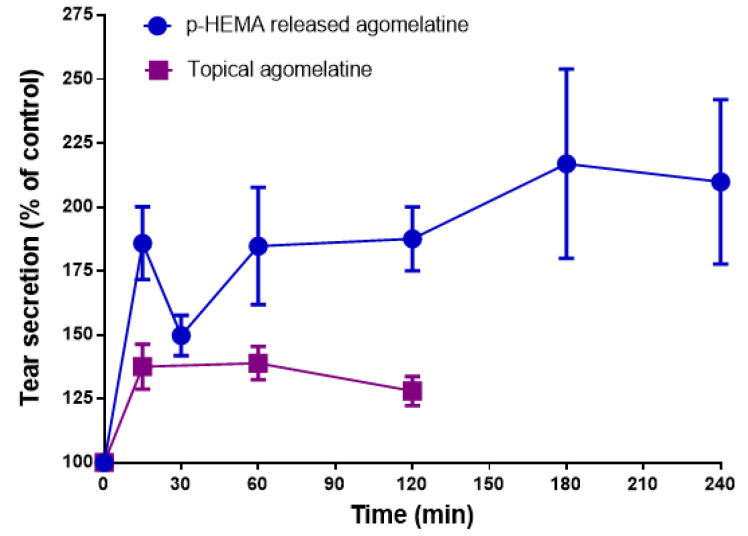
The effect of agomelatine released from CLs and administered topically in tear secretion. The P-HEMA lenses were preloaded with agomelatine 100 μM for 12 h and 10 μL of agomelatine 100 μM was administered topically on rabbit eyes. Tear secretion was evaluated over 240 min (for CLs experiments) and over 180 min (for topical experiments), versus the basal tear secretion and tear secretion after topical application of a saline solution containing 1% DMSO, respectively (100% tear secretion). Tear secretion values (%) are presented as the mean ± S.E.M. of *n* = 12 lenses and *n* = 24 rabbit eyes.

**Figure 8 jcm-11-03483-f008:**
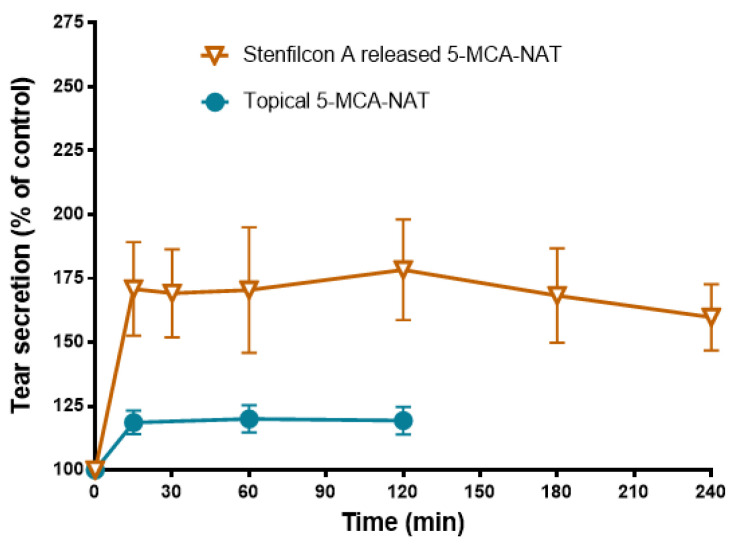
The effect of 5-MCA-NAT released from CLs and administered topically in tear secretion. Stenfilcon A lenses were preloaded with 5-MCA-NAT 250 μM for 12 h and 10 μL of 5-MCA-NAT 250 μM was administered topically on the rabbit eyes. Tear secretion was evaluated over 240 min (for CLs experiments) and over 180 min (for topical experiments) versus the basal tear secretion and tear secretion after the topical application of a saline solution containing 1% DMSO, respectively (100% tear secretion). The tear secretion values (%) are presented as mean ± S.E.M. of *n* = 12 lenses and *n* = 24 rabbit eyes.

**Table 1 jcm-11-03483-t001:** The properties of the silicone and conventional hydrogel contact lenses used in the study.

Trade Name	PureVision	Biofinity	MyDay	Proclear	Veraflex
Unit states adopted name (USAN)	Balafilcon A	Comfilcon A	Stenfilcon A	Omafilcon B	p-HEMA
Manufacturer	Bausch & Lomb	Coopervision	Coopervision	Coopervision	Interlenco
Center thickness	0.07	0.08	0.08	0.07	0.08
Water Content (%)	36	48	54	62	38
Oxygen Permeability (×10^−11^)	91	128	80	27	NA
Oxygen Transmissibility (×10^−11^)	130	160	100	42	30
FDA Group	III	I	II	II	I
Surface Treatment	Plasma oxidation process	None	None	None	None
Principal monomers	NVA, NVP, PBVC, TPVC	FM0411M, HOB, IBM, M3U, NVP, TAIC, VMA	EGDMA, EGMA, NB, PDMS, PMMA, TEGDVE, VMA	EGDMA, HEMA, MPC	HEMA, NVP

EGDMA, ethylene glycol dimethacrylate; EGMA, ethylene glycol methyl ether methacrylate; FM0411M, α-methacryloyloxyethyl imninocarboxyethyloxypropylpoly (dimethylsiloxy)-butyldimethylsilane; HEMA, hydroxyethyl methacrylate; HOB, 2-hydroxybutyl methacrylate; IBM, isobornyl methacrylate; MPC, 2-methacryloyloxyethyl phosphorylcholine; M3U, α-ω-bis(methacryloyloxyethyliminocarboxyethyloxypropyl)-poly(dimethylsiloxane)-poly(trifluoropropylmethylsiloxane)-poly(v-methoxy-poly(ethyleneglycol)propyl methylsiloxane; NA, not available; NB, norbloc, 2-[3-(2H-Benzotriazol-2-yl)-4-hydroxyphenyl]ethyl methacrylate; NVA, N-Vinyl Ala; NVP, N-Vinyl pyrrolidone; PBVC, poly(dimethysiloxy) di(silylbutanol) bis(vinyl carbamate; PDMS, polydimethylsiloxane; PMMA, Poly methyl methacrylate; TAIC, 1,3,5-triallyl-1,3,5-triazine-2,4,6(1H,3H,5H)-trione; TEGDVE, triethylene glycol divinyl ether; TPVC, tris-(trimethylsiloxy) silyl propyl vinyl carbamate; VMA, N-vinyl-N-methylacetamide.

**Table 2 jcm-11-03483-t002:** The yield of contact lenses loaded with agomelatine as a drug delivery system.

Unit States Adopted Name (USAN)	Maximum Absorbed Concentration (μM)	Maximum Released Concentration (μM)	μM Ratio
p-HEMA	613.97 ± 21.36	166.60 ± 3.97	3.7
Omafilcon B	691.67 ± 19.64	151.07 ± 14.43	4.6
Comfilcon A	884.7 ± 41.42	140.43 ± 17.45	6.3
Stenfilcon A	906.44 ± 19.56	138.25 ± 12.37	6.6
Balafilcon A	895.43 ± 11.45	101.04 ± 11.52	8.9

**Table 3 jcm-11-03483-t003:** The yield of contact lenses loaded with 5-MCA-NAT as a drug delivery system.

Unit States Adopted Name (USAN)	Maximum Absorbed Concentration (μM)	Maximum Released Concentration (μM)	μM Ratio
p-HEMA	507.98 ± 20.14	591.38 ± 15.46	0.9
Omafilcon B	407.55 ± 26.88	521.38 ± 69.58	0.8
Comfilcon A	648.00 ± 38.21	747.89 ± 32.24	0.9
Stenfilcon A	681.44 ± 23.68	781.88 ± 25.13	0.9
Balafilcon A	397.04 ± 15.86	728.95 ± 36.04	0.5

## Data Availability

Not applicable.

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
