# Peer review of "Contact Lenses Loaded with Melatonin Analogs: A Promising Therapeutic Tool against Dry Eye Disease"

_jcm, 2022, doi:10.3390/jcm11123483_

Round 1

Reviewer 1 Report

General comments:

It was interesting reading the reported study. I congratulate the authors for the innovative ideas of this study. Furthermore, the study was conducted in a precise and sustainable manner. 

Specific comments

Line 155: The Schirmer test is non-specific and varies between individuals. Just so I understand better, it is correct that each rabbit was given a CL without melatonin before the processed CL with melatonin was applied. If so, then that should be more clearly stated. With a completely uncoupled sample group, comparing the Schirmer test becomes more difficult.

Line 170: The Shapiro-Wilks test is very useful to test the distribution of the samples. How were the samples that did not show a normal distribution further processed?

Line 424: Can the control group be precisely defined. As already mentioned above, the Schirmer test is not consistent interindividually and thus the absolute values are not always easy to compare. This should also be mentioned.

Line 588: It was rightly mentioned that the clinical results from rabbits cannot simply be transferred to humans. However, this is often the first step in obtaining a reliable evaluation. However, some questions should be mentioned in the discussion. What is the potential clinical hazard of melatonin at this concentration on corneal epithelial cells? What is the biocompatibility of melatonin on CL? Are allergies to be expected? Is trophic disturbance of the corneal tissue to be expected? Another important point is what would a clinical trial in humans look like? How long would the CLs have to be worn?

Line 598: It should be mentioned that melatonin treatment tends to improve the aqueous phase of the tear film. This would certainly help patients with Sjögren's syndrome. However, most dry eye patients suffer from meibomian gland dysfunction.

Reviewer 2 Report

The paper “Contact Lenses loaded with melatonin analogs: a promising 2 therapeutic tool against Dry Eye Disease”, reported an interesting topic about  the in vitro ability of five commercially available hydrogels CLs to act as a delivery system for melatonin analogs and the in vivo secretagogue effect of melatonin analogs-loaded CLs.

I agree with the limitation of this study regarding the the eye toxicity of the compounds of melatonin and thus its analogs in eye diseases.

Only few specific concerns: 

Please add in the introduction section more informations about the role of Melatonin  as a neurohormone involved in the regulation of tear and aqueous humor production and the associated physiological processes.

Lines 80-84 of the introduction section are part of the results/discussion section. Therefore, please remove this part from introduction and add it in the results section. In the final part of introduction please add the aim of the study.

Lines 580-581 lines: Please add more information about the possible side effect. 

Please add a previous reference https://doi.org/10.3390/pharmaceutics14051019
